elifesciences.org eLIFE

# Nonenzymatic copying of RNA templates containing all four letters is catalyzed by activated oligonucleotides

**Noam Prywes[1,2], J Craig Blain[2†], Francesca Del Frate[2], Jack W Szostak[1,2]***

[1]Department of Chemistry and Chemical Biology, Harvard University, Cambridge, United States; [2]Department of Molecular Biology and Center for Computational and Integrative Biology, Howard Hughes Medical Institute, Massachusetts General Hospital, Boston, United States

**Abstract** The nonenzymatic replication of RNA is a potential transitional stage between the prebiotic chemistry of nucleotide synthesis and the canonical RNA world in which RNA enzymes (ribozymes) catalyze replication of the RNA genomes of primordial cells. However, the plausibility of nonenzymatic RNA replication is undercut by the lack of a protocell-compatible chemical system capable of copying RNA templates containing all four nucleotides. We show that short 5′-activated oligonucleotides act as catalysts that accelerate primer extension, and allow for the one-pot copying of mixed sequence RNA templates. The fidelity of the primer extension products resulting from the sequential addition of activated monomers, when catalyzed by activated oligomers, is sufficient to sustain a genome long enough to encode active ribozymes. Finally, by immobilizing the primer and template on a bead and adding individual monomers in sequence, we synthesize a significant part of an active hammerhead ribozyme, forging a link between nonenzymatic polymerization and the RNA world.

**\*For correspondence:** szostak@ molbio.mgh.harvard.edu

**Present address:** [†]Ra Pharmaceuticals, Cambridge, United States

**Competing interests:** The authors declare that no competing interests exist.

## Introduction

In order to serve as a bridge from prebiotic chemistry to the RNA world (*Patel et al., 2015*; *Powner et al., 2009*), the nonenzymatic copying of RNA templates by suitably activated nucleotides (*Figure 1a*) must occur quickly enough to replicate functional RNA sequences faster than they degrade. However, the nonenzymatic copying of sequences containing all four nucleotides has not yet been possible for a number of reasons (*Szostak, 2012*), most notably the slow rate of primer extension with adenosine and uridine monomers (*Joyce et al., 1987*; *Wu and Orgel, 1992b*; *Deck et al., 2011*). Indeed, two adjacent A or U residues in the template will stop polymerization entirely (*Wu and Orgel, 1992b*) under normal conditions; even partial copying requires extreme conditions, such as the eutectic phase of frozen samples (*Vogel and Richert, 2007*), which are not compatible with replication within protocells. Extensive optimization of reaction conditions with the aim of improving the rate of primer extension with all four monomers, including varying the choice of divalent cation and leaving group, improved the rate and regiospecificity of polymerization for G and C but not A and U (*Wu and Orgel, 1992b*, *c*; *Deck et al., 2011*; *Inoue and Orgel, 1981*; *Hagenbuch et al., 2005*; *Lohrmann et al., 1980*). Recent biophysical studies suggest that low A and U monomer affinity for the template is not the only reason for this problem (*Szostak, 2012*; *Izgu et al., 2015*) since A binds only three-fold more weakly than G, but primer extension with A is at least 100-fold slower (*Joyce et al., 1987*; *Wu and Orgel, 1992b*; *Deck et al., 2011*; *Heuberger et al., 2015*) than with G. Moreover, Deck et al. have shown that downstream helper oligonucleotides (*Deck et al., 2011*) that provide incoming monomers with an additional binding

**eLife digest** Though defining what makes something "alive" has proved challenging, one crucial feature of living things is the ability to copy genetic information and pass it on to the next generation. Nowadays, enzymes called polymerases copy genetic information encoded within the DNA of living cells. However, when life on Earth began approximately four billion years ago, polymerases had not evolved yet. This means that the genetic information of the first cells had to be copied some other way.

The earliest life on Earth is unlikely to have used DNA to store its genetic information, and probably used a closely related molecule called RNA instead. Like DNA, RNA is made up of four smaller building blocks joined together to form long chains. The building blocks of RNA are commonly referred to using single letters: A, C, G and U. Previous studies have shown that it is possible to copy RNA without enzymes, but for only two of the four RNA letters, namely C and G.

Prywes et al. wanted to know if it was possible to create a chemical system, without polymerases, in which all four RNA letters could be copied. The experiments showed that strings of RNA that were three letters long could catalyze RNA copying, just as long as they were chemically activated. That is to say, these short RNA strings allowed RNA to be copied without enzymes if they had a chemical group at one end that made them more reactive.Each short catalyst helped copy one of the four RNA letters, and adding several into one reaction meant that longer sequences containing all four RNA letters could be copied.

Prywes et al. then used these short catalysts to copy an RNA molecule that itself acts a bit like an enzyme, and confirmed that a significant portion of this molecule could be copied without any polymerases. Further work is now needed to see if it is possible to copy other RNA sequences, and especially longer ones, without enzymes. Another challenge for the future would be to attempt to copy an RNA sequence multiple times without enzymes; a challenge that the earliest ancestors of cells on Earth must have overcome to pass their genetic information down through the generations.

surface improve the binding of A and U monomers, but the underlying difference in rates remains. The use of helper oligonucleotides allowed primer extension to proceed on a template containing all four bases over the course of several weeks, where previously such templates could not be copied at all.

Here we show that the presence of a leaving group at the 5′ end of a helper oligonucleotide can be a source of significant catalysis. Activated helper oligonucleotides facilitate the addition of all four RNA monomers individually and in sequence in a single reaction. Finally, when a template is immobilized by attachment to a bead, sequential addition of activated monomers and helper oligonucleotides successfully promoted the synthesis of a significant part of an active hammerhead ribozyme.

## Results

Given the historical and continuing problems in demonstrating efficient nonenzymatic template directed primer extension, one might wonder whether this biologically inspired model is appropriate for replication within primordial cells. An alternative scenario that might seem more reasonable in the context of prebiotic chemistry involves template copying by the initial formation of short oligonucleotides, followed by ligation events that generate longer oligonucleotide intermediates and eventually a full length copy (*Szostak, 2011*; *James and Ellington, 1999*). To test the viability of this hierarchical assembly model, we compared the rate of primer extension with activated monomers to the rate of ligation of two template bound oligonucleotides (*Figure 1b,c* reactions 1 and 3). Nucleotides activated with 2-methylimidzole on the 5′-phosphate have been used extensively to model nonenzymatic template copying (*Deck et al., 2011*; *Wu and Orgel, 1992c*), irrespective of their prebiotic plausibility, and were used in all experiments reported here. In both the nucleotide polymerization and the oligonucleotide ligation reactions, the chemical reaction is the same, i.e. attack of the 3′-hydroxyl of an oligonucleotide on the 5′-phosphate of a downstream nucleotide, with displacement of the 2-methylimidazole leaving group (*Figure 1b*). Moreover, the reacting nucleotides

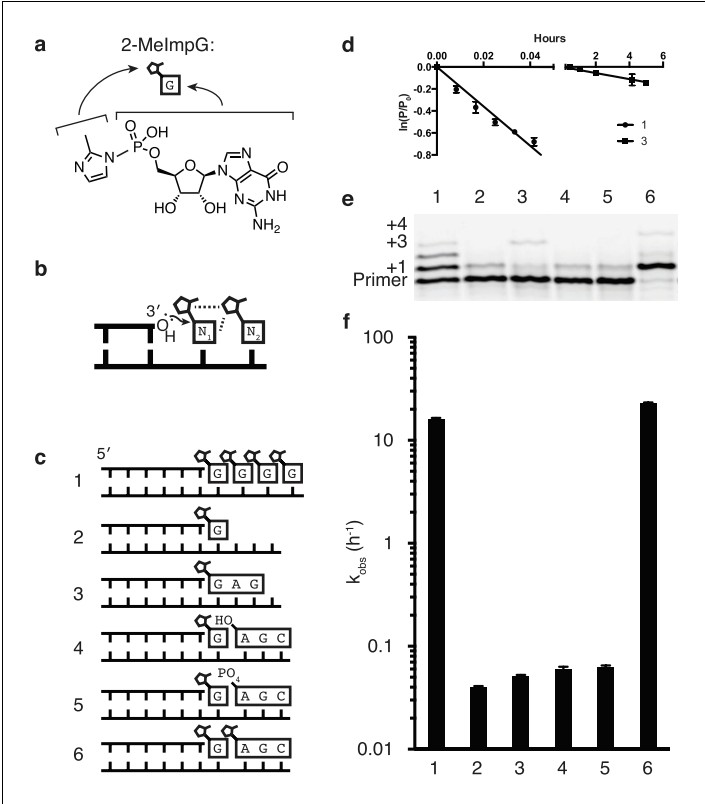

**Figure 1.** Catalysis of nonenzymatic primer extension by activated downstream nucleotides. (**a**) Structure of 2-methylimidazole-activated guanosine-5'-monophosphate, and its schematic representation. (**b**) Schematic of the RNA primer extension reaction. $N_1$ represents an individual ribonucleotide monomer in position to react with the primer, $N_2$ represents either a monomer or an oligomer downstream, with a leaving group capable of interacting with $N_1$. Dashed lines: Potential interactions between leaving groups or between the downstream leaving group and the upstream nucleotide $N_1$. (**c**) Schematics of the RNA primers, templates, monomers and oligomers used in **d–f**. Templates are complementary to the displayed monomers and oligomers. Template 2 has a U following the C to which the G monomer is bound to prevent downstream binding of G. (**d**) Primer extension by polymerization or ligation on templates 1 or 3, respectively. Fits describe ln(fraction primer remaining) vs time, giving an apparent first-order rate constant (*Figure 1—figure supplement 1*). (**e**) Primer extension assay for the experiments described in **c**, showing reaction progress after 10 min. In lane 6, a primer + 4 band can be observed, representing the slow addition of the activated trimer after the monomer. (**f**) Pseudo-first order rates of the reactions described in **c**. Error bars indicate S.E.M; all experiments were performed in triplicate or greater. Reaction conditions: 10 μM primer, 11 μM template, 200 mM CHES pH 9.0, 200 mM $MgCl_2$, 50 mM monomer, 1 mM trimer.

The following figure supplements are available for figure 1:

**Figure supplement 1.** Determination of the rates of primer extension reactions.

**Figure supplement 2.** Rates of additional trimer ligations.

are the same in both cases: a G at the 3'-end of the primer and a G monomer or a G at the 5'-end of an oligonucleotide. We expected the ligation reaction to be faster, because the oligonucleotides to be ligated are both stably bound to the template, and because the fully base-paired nicked duplex should be largely in the optimal A-type conformation (*Kozlov and Orgel, 2000*), whereas the primer/template complex with template bound monomers would be significantly more disordered. To our surprise, the primer extension reaction proceeded almost 100 times more rapidly than the ligation reaction (*Figure 1d*). This was true for a variety of trimer sequences (*Figure 1—figure supplement 2*). In our search for an explanation for this unexpected observation, we reasoned that one of the major differences between the two scenarios was the presence of multiple adjacent 5'-

activated nucleotides downstream of the primer in the case of monomer addition, but not ligation, where there is only a single activated nucleotide. Indeed, the case for a role of a second downstream activated nucleotide is consistent with the well known difficulty of extending a primer to the last nucleotide of a template (*Wu and Orgel, 1992c*). Furthermore, Orgel et al. showed over 20 years ago that efficient primer extension requires the presence of two activated monomers adjacent to the primer. Remarkably, the reaction is fastest when both monomers are activated with 2-methylimidazole instead of either or both being activated with imidazole (*Wu and Orgel, 1992a*), suggesting a possible catalytic role for a physical interaction of the leaving groups of adjacent monomers.

In order to directly test the possibility that a 2-methylimidazole leaving group on the 5′-phosphate of a downstream monomer or oligonucleotide (*Figure 1b*) would increase the rate of reaction between a primer and an adjacent monomer, we designed a series of templates for nonenzymatic RNA polymerization and ligation (*Figure 1c*). We then measured the rate of addition of a monomer to a primer in the presence or absence of downstream nucleotides, with or without 5′-activation of the downstream nucleotides (*Figure 1c–f*). Rates were calculated assuming a pseudo-first order rate equation (*Figure 1—figure supplement 1*), because the concentrations of primer and template were far below that of the monomer, which would not change significantly during the reaction. Primer extension by addition of a single monomer, in the absence of any additional downstream mono- or oligo-nucleotides, was very slow and comparable in rate to the ligation of the primer to a 5′-activated GAG trinucleotide (*Figure 1c–f*, reactions 2 and 3). The binding of an unactivated AGC trimer, with either a 5′ hydroxyl or a 5′ phosphate, downstream of an activated G monomer conferred only a modest increase in the rate of addition of the G monomer to the primer (*Figure 1c–f*, reactions 4 and 5), consistent with previous reports of unactivated downstream 'helper' oligonucleotides (*Deck et al., 2011*; *Jauker et al., 2015*; *Kervio et al., 2010*). Such helper oligonucleotides are thought to act largely by increasing monomer affinity to the primer/template complex by providing an additional base-stacking surface. In contrast, when the same AGC trimer was activated as a 5′-phosphoro-2-methylimidazolide and used as a downstream helper oligonucleotide, the rate of primer extension (by addition of a G monomer) was increased by over two orders of magnitude (*Figure 1c–f*, reaction 6) compared to reactions with unactivated trimers. The rate of primer extension with a single G in the presence of an activated downstream trinucleotide was about twice as fast as the rate when the primer was followed by up to four sequential activated G monomers (*Figure 1c–f*, reaction 1 and 6).

To systematically study the rate of nonenzymatic primer extension as a function of the length of the downstream activated 'helper' oligonucleotide, we synthesized a series of activated oligonucleotides of different lengths (*Figure 2a*). At saturating concentrations (*Figure 2—figure supplement 1*) of each of the oligonucleotides, the rate of primer extension improved as the length of the helper increased from mononucleotide to dinucleotide to trinucleotide, and then stayed approximately constant with the tetranucleotide (*Figure 2a*). At subsaturating concentrations of helper oligonucleotide, the rate of the primer extension reaction increased with increasing concentrations of di- or tri-nucleotide helper (*Figure 2—figure supplement 1*). Because the maximum rate of the trimer-assisted reaction was significantly faster than that of the dimer-assisted reaction, we proceeded with trimers as the downstream helper oligonucleotides for the remainder of this study.

We then tested the ability of a downstream 2-methylimidazole-activated AGC trimer to catalyze template-directed RNA primer extension with all four individual 2-methylimidazole activated monomers – A, G, C and U. We also tested the 2-thiouridine monomer, which we have previously shown to be superior to uridine in nonenzymatic primer extension in terms of both increased rate and improved fidelity (*Heuberger et al., 2015*). In each case, the rate of primer extension increased by at least two orders of magnitude compared to the corresponding reactions without any helper oligonucleotides (*Figure 2b*). Remarkably, whereas the rate of primer extension with A and U monomers was previously too slow to measure (*Wu and Orgel, 1992b*; *Heuberger et al., 2015*) – conservatively estimated to be at least three orders of magnitude slower than primer extension with G or C monomers – the difference in polymerization rates was reduced to roughly one order of magnitude when assisted by downstream activated trimers (*Figure 2b*).

In order to assess the fidelity of nonenzymatic trimer-assisted primer extension, we measured the rates of monomer addition for all monomer-template combinations, including both matches and mismatches (*Figure 3a*). We calculated the fidelity for each template as the rate of matched monomer addition divided by the sum of matched and mismatched monomer addition rates. By averaging the

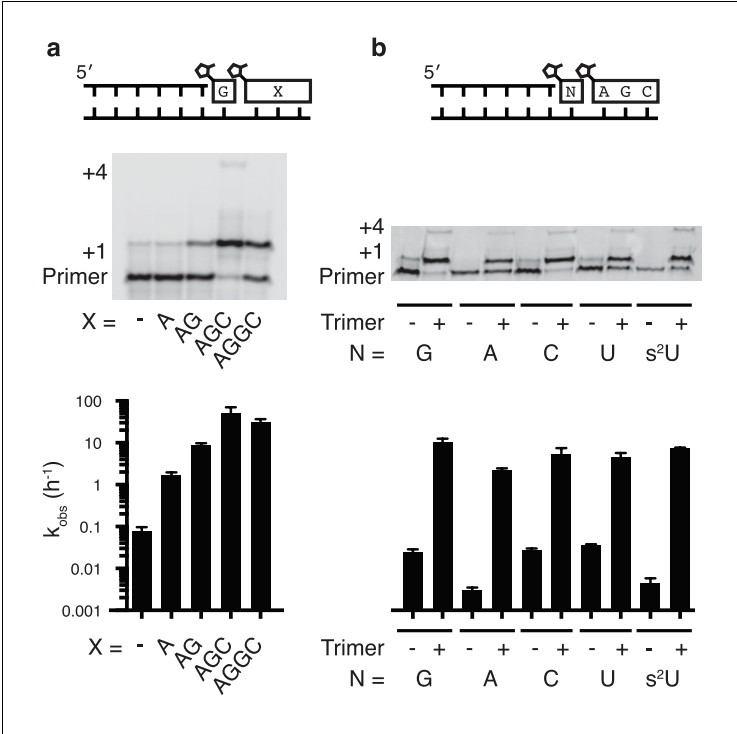

**Figure 2.** Nonenzymatic primer extension using all four monomers. (**a**) RNA primer extension assay using alternative 'helper' oligomers and corresponding, complementary templates. The respective rates are shown at bottom. Concentration curves are provided for AG and AGC in *Figure 2—figure supplement 1*. (**b**) Primer extension assay for each of the four monomers, as well as 2-thiouridine, in the absence and presence of activated trimer. Shown at bottom of both **a** and **b** is a bar graph with the respective rates. Reaction conditions: 10 μM primer, 11 μM template, 200 mM Tris pH 8.0, 100 mM MgCl$_2$, 50 mM monomer, 10 mM 2-MeImpAG, 1 mM 2-MeImpAGC, 1 mM 2-MeImpAGGC. The gels in **a** and **b** show reaction progress after 10 min and 1 hr respectively. Error bars indicate S.E.M; all experiments were performed in triplicate or greater.

The following figure supplement is available for figure 2:

**Figure supplement 1.** Rate of primer extension vs. concentration of downstream activated di- and tri-nucleotides.

---

fidelities of the four templates we found the overall fidelity of trimer-assisted polymerization to be 98%. Assuming an error threshold of one mutation per genome (*Eigen, 1971*), this value of fidelity allows for effective genome sizes of 50 nucleotides, long enough to produce functional ribozymes (*Ferré-D'Amaré and Scott, 2010*). The fastest mismatch reaction was primer extension by G during copying of a template U, which was approximately 5% as fast as primer extension with A on the same template (*Figure 3a*, *Figure 3—figure supplement 1*). This is likely due to the formation of a G:U wobble base pair (*Heuberger et al., 2015*), and is consistent with previous studies of nonenzymatic primer extension reactions (*Rajamani et al., 2010*). All other mismatches were at least two orders of magnitude slower than their matched counterparts. We also tested the effect of replacing the canonical U monomer with 2-thiouridine and 2-thioribothymine on the polymerization rate and found, in line with previous reports (*Heuberger et al., 2015*), that this sulfur substitution improved the rates by approximately one order of magnitude (*Figure 3b*), bringing the rate of primer extension closer to the rates of G and C addition. More rigorous tests of fidelity, in which monomers can compete for binding on a template in all possible sequence contexts, are ongoing.

Encouraged by the relatively fast and accurate addition of all four monomers to a primer, we then attempted to copy short RNA templates containing all four nucleotides. In order to iterate the process of trimer assisted monomer addition, after a downstream activated trimer catalyzes the addition of the first monomer to the primer, the helper trimer must dissociate to allow for the binding of the next monomer-trimer pair. To test the feasibility of this mode of primer extension, we designed a

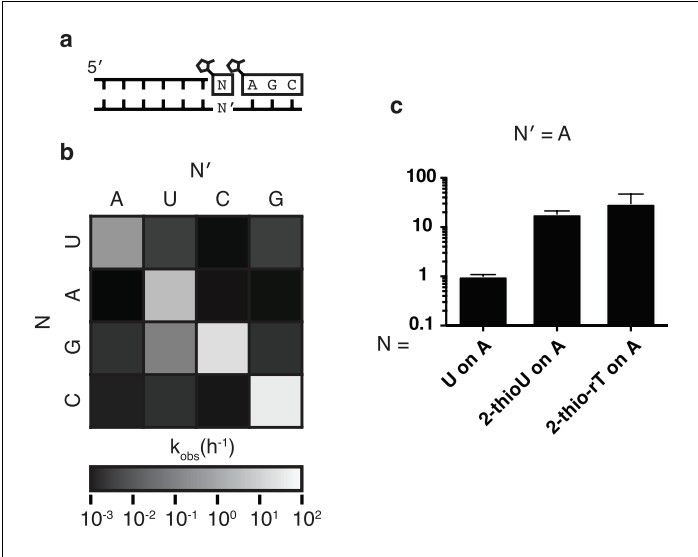

**Figure 3.** Fidelity of trimer-assisted primer extension. (**a**) Schematic of the fidelity assay. Using the same RNA primer and trimer, four different templates, one with each base, were paired with each of the four 2-methylimidazole activated monomers to test the relative rates of each matched and mismatched pair. (**b**) Heat map showing the relative rates of primer extension for each template:monomer pairing. Black indicates a rate of $0.001 \, h^{-1}$, white indicates a rate of $100 \, h^{-1}$. A bar graph and a table of the same data can be found in **Figure 3—figure supplement 1**. (**c**) Relative rates of primer extension with the modified U monomers, 2-thiouridine and 2-thioribothymidine compared to that of a canonical, activated U monomer on a template containing an A. Reaction conditions as in **Figure 2**. Error bars indicate S.E.M. All experiments were performed in triplicate or greater.

The following figure supplement is available for figure 3:

**Figure supplement 1.** Rates of trimer-assisted primer extension for all monomer/template combinations.

template that contained binding sites for all four monomers and synthesized the appropriate activated trimers (**Figure 4a**). We used 2-thiouridine monomer in place of uridine (see above), and all four monomers (A, G, C and 2-thiouridine) were present in every reaction. In the absence of activated trimers, we observed almost no primer extension (lane 1 **Figure 4b**); even though all four monomers were present, their combined rate of addition was not measurable. Adding just the first helper trimer resulted in rapid addition of the first monomer (C) to the primer, followed by slow addition of the first trimer (AGC) to the growing primer. Similarly, adding two, three or four trimers together resulted in the generation of primers extended by two, three or four monomers respectively (**Figure 4b**). These reactions converted primer into a product with four sequentially added monomers with an ~80% yield after 16 hr (**Figure 4c,d**); single monomer addition reactions reached >95% yield after only 10 min (**Figure 2b**). The difference between the fast rates observed in the trimer-assisted addition of a single base (**Figure 2b**) and the slower rate of trimer-assisted polymerization of all four bases simultaneously (**Figure 4b**) may be due to the competition of trimers for overlapping binding sites on the template. This hypothesis is supported by the decreased rate of trimer assisted primer extension with a single monomer in the presence of additional overlapping downstream trimers (**Figure 4—figure supplement 1**). The copying of templates containing two consecutive A or U monomers is also made possible by the use of activated helper oligonucleotides (**Figure 5**).

If nonenzymatic RNA polymerization did indeed precede the RNA world, then it would have been necessary to synthesize RNA sequences long enough to function as ribozymes (**Deck et al., 2011**). While our method can only produce sequences >7 nucleotides in one reaction in low yields (**Figure 6—figure supplement 1**) due to the cross-inhibitory effect of trimers with overlapping binding sites (**Figure 4—figure supplement 1**), we reasoned that a ribozyme could be synthesized by this approach if monomers were added, along with their attendant trimers, one at a time (**Figure 6**). To

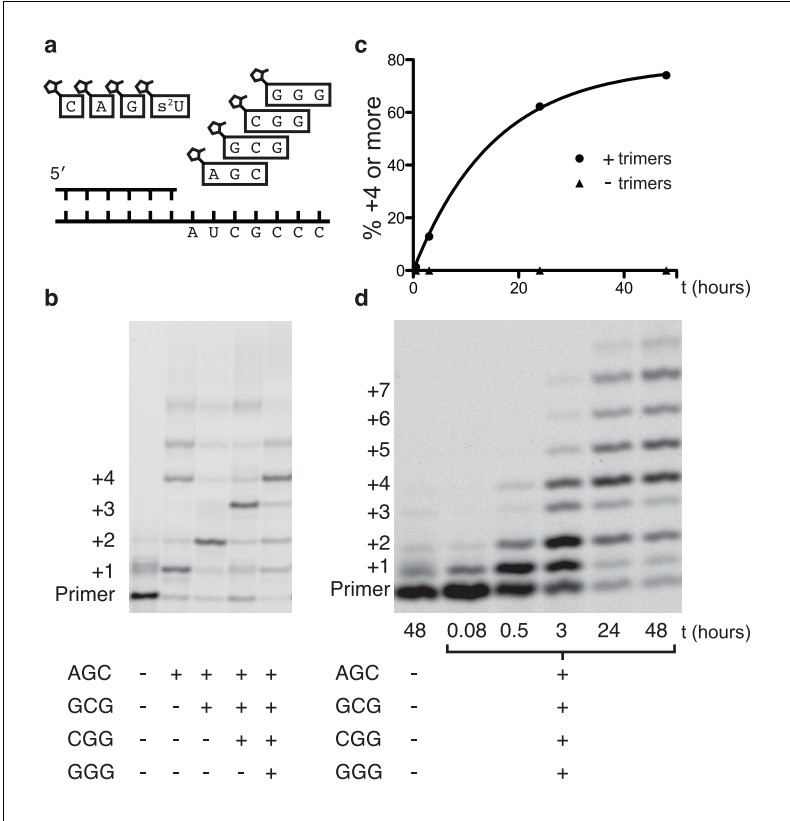

**Figure 4.** Primer extension with all four monomers in a one-pot reaction. (**a**) Schematic of a primer extension reaction incorporating all four RNA monomers, with four respective downstream helper trimers. (**b**) All four monomers and the activated trimers indicated below the gel were added to each reaction. For example, in lane 3 all four monomers and the activated AGC and GCG trimers were mixed with primer, template, buffer and Mg²⁺. Higher bands represent either the ligation of activated trimers or the polymerization of G monomers without trimer assistance. The gel shows reaction progress after 16 hr. (**c**) Timecourse of the reaction with all four trimers. A 48 hr timepoint of the reaction without any trimers (left lane of b) is contrasted with a timecourse (timepoints at 5 and 30 min, 2, 24 and 48 hr). The percentage of polymerization products that have been extended by at least four nucleotides is plotted over time. (**d**) A gel displaying the timepoints plotted in part c. A timecourse showing the effect of overlapping trimers on reaction rate can be found in *Figure 4—figure supplement 1*. Reaction conditions: 2.5 µM primer, 2.5 µM template, 100 mM HEPES pH 8.0, 100 mM MgCl₂, 20 mM monomer, 100 µM trimer.

The following figure supplement is available for figure 4:

**Figure supplement 1.** Inhibition of primer extension by overlapping helper-trimers.

that end, we immobilized a template that codes for the sequence of one half of the hammerhead ribozyme (HH2) – on a bead and added matched pairs of monomers and trimers sequentially (*Figure 6a*). Primer extension was monitored by gel electrophoresis. Despite the fact that no 2-thio-uridine monomers were used (in order to protect the catalytic capacity of the ribozyme) a high conversion rate was observed at each step (75–95%) (*Figure 6b*). The reaction durations required to achieve these conversion rates varied widely, for example, the seventh monomer addition required 21 hr to reach 95% while the eighth addition reached the same level of conversion after just 30 min. Generally, G and C additions were much faster than A or U additions; the reason for the discrepancy between the rates of these reactions on beads and those in solution is unknown. The use of s²U or s²T might offset some of the discrepancy. The final product was released from the bead by hybridization of an excess of a DNA oligonucleotide of identical sequence to the immobilized template. After DNase treatment, the hammerhead substrate, HH1, was added to the crude synthesized pool

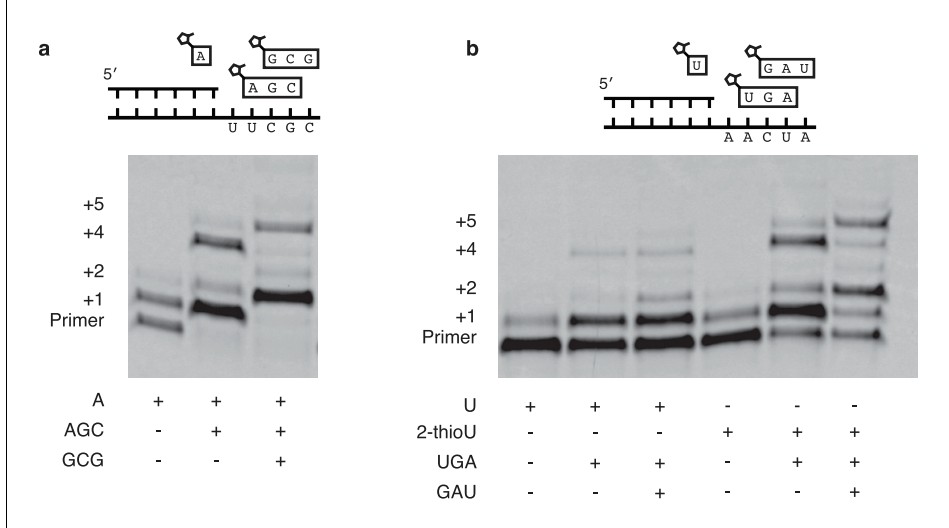

**Figure 5.** Extension of primers by multiple consecutive A or U nucleotides. (a) Schematic of a primer extension reaction wherein two consecutive A nucleotides can polymerize. The gel shows extension without trimers and with one or two trimers after 16 hr. (b) Same as in (a) but with two U additions. In the right half 2-MeImpU is replaced with 2-MeImp-2-thioU.

of oligonucleotides in the presence of 200 mM MgCl$_2$ for 16 hr (*Figure 6c*). Despite the fact that this was a crude reaction product, with full-length HH2 comprising only 6% of the reaction products, the product mixture was able to function as a catalyst, with 50% ribozyme cleavage yield, thus demonstrating that a significant portion of an active ribozyme was synthesized nonenzymatically.

## Discussion

The experiments described above demonstrate for the first time a simple and robust means of nonenzymatically copying mixed sequence RNA templates. Short, activated oligonucleotides – themselves plausibly generated by either templated or untemplated monomer polymerization – are efficient catalysts of high fidelity primer extension with all four RNA monomers. Thus short activated oligonucleotides can play dual roles, acting either as catalysts of chain growth by monomer addition, or directly as replication intermediates, since they can also be incorporated into a growing chain by ligation. Template copying in a complex but realistic milieu containing both activated monomers and oligomers could therefore occur via a hybrid process combining primer extension with monomers and oligonucleotide ligation.

The disparity in rates between the reactions catalyzed by activated vs. unactivated trimers indicates a key role for the leaving group downstream of the polymerizing monomer. The slow rate of activated oligonucleotide ligation is also explained by the lack of a leaving group one base downstream of the site of the primer extension reaction: in this case the downstream position corresponds to the second monomer in the oligonucleotide which necessarily has no leaving group as it is linked to the upstream monomer by a phosphodiester linkage. The nature of the interaction between the downstream leaving group and the reacting monomer remains unclear. It is possible that the downstream leaving group interacts attractively with the reacting leaving group, positioning it optimally for attack by the 3′ hydroxyl of the primer. As observed by Wu et al. (*Wu and Orgel, 1992a*) this interaction could be very sensitive to steric effects, such that 2-methylimidazole at the downstream position would facilitate polymerization much more effectively than imidazole, though the authors ultimately concluded that such subtle effects are not likely to adequately explain the strength of the interaction. Wu et al. (*Wu and Orgel, 1992a*) also suggested that the catalytic effect could be due to acid-base catalysis. Another possibility is that the upstream leaving group forms a covalent bond with the phosphate of the downstream monomer, a reaction that is only possible if there is a leaving group on the downstream monomer (*Kervio et al., 2016* and T. Walton and J.W. Szostak,

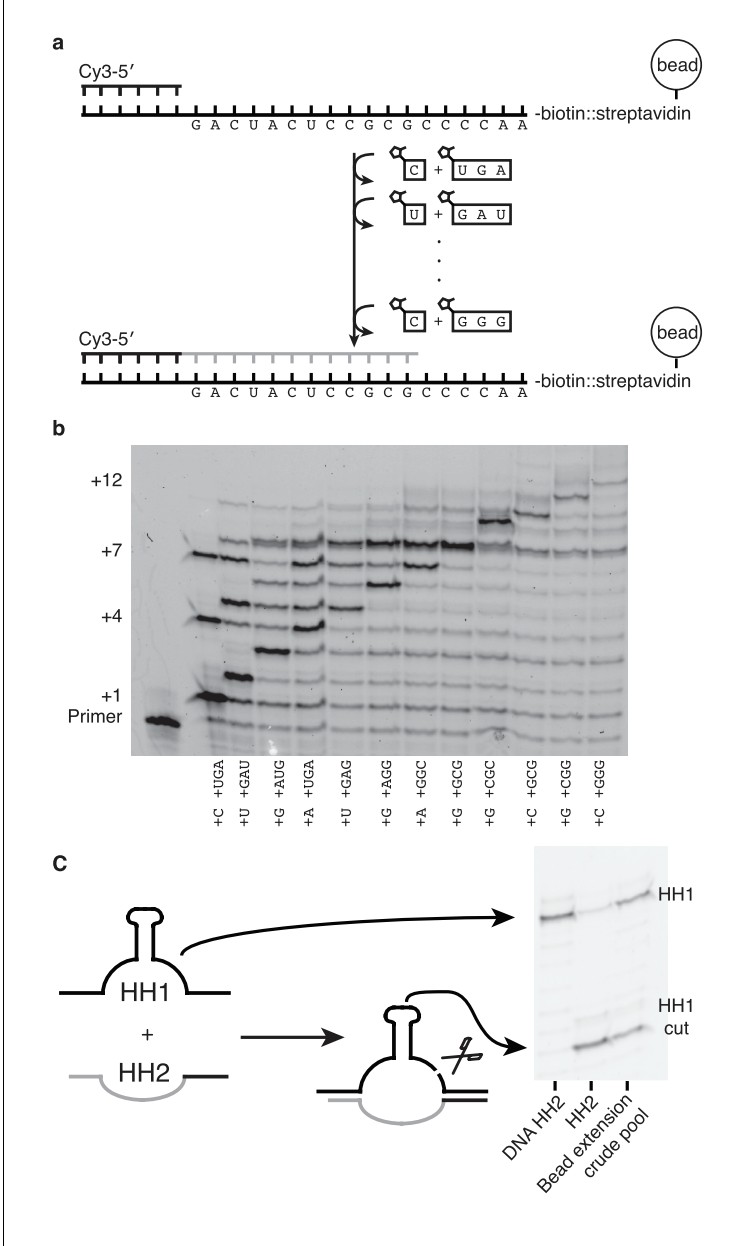

**Figure 6.** Nonenzymatic RNA synthesis of an active hammerhead ribozyme by template dependent primer extension. (a) Schematic of the bead-assisted primer extension reaction. Synthesis of the hammerhead ribozyme strand HH2 by the sequential addition of matched pairs of activated monomers and trimers. (b) Each lane shows primer extension by one nucleotide, catalyzed by the corresponding downstream activated trimer. Final yield of the +12 product was 6%. Reaction conditions: 100 mM MgCl$_2$, 200 mM Tris-HCl pH 8, 25 mM monomer and 2 mM trimer. The beads were washed between steps in 2 M NaCl, 1 mM EDTA and 10 mM Tris-HCl pH 7. For steps 2, 5 and 7, 50 mM monomer and 4 mM trimer were added. Each step was monitored by gel electrophoresis until it reached >75% yield. Steps varied in length from 30 min to 22 hr; the total time required was 150 hr. (c) The product of the last bead-assisted reaction was removed from the bead by the addition of the an excess of the DNA sequence of the extended primer at 95°C over 2 min. After DNase treatment, the target hammerhead RNA, HH1, was added. The third lane shows the extent of the nuclease reaction, 50%, after 16 hr at room temperature. For comparison, pure HH2 made from DNA and RNA served as negative and positive controls respectively. Reaction conditions: Extended primer from the beads was added to 200 mM MgCl$_2$, 200 mM Tris-HCl pH 8 and 250 nM Cy5 labeled HH1 RNA.

The following figure supplement is available for figure 6:

*Figure 6 continued on next page*

*Figure 6 continued*

**Figure supplement 1.** One pot synthesis of hammerhead ribozyme.

unpublished data). This covalent bond could form before the attack of the 3′ hydroxyl of the primer, creating an imidazolium intermediate, or in a concerted reaction where the leaving group of the upstream monomer replaces the leaving group of the downstream monomer. Further understanding of the reaction mechanism could lead to concomitant improvements in the rate and extent of template copying.

Further advances are required to enable the copying of longer and potentially functional RNA sequences in one pot, protocell-compatible conditions. However, the copying of long mixed template sequences under our conditions compares favorably, both in terms of rate and efficiency with reactions in which arbitrary RNA templates are copied by highly evolved ribozyme polymerases (*Wochner et al., 2011*) (*Figure 6—figure supplement 1*). In fact, if the rates and fidelities of nonenzymatic RNA polymerization were sufficiently high, the need for a ribozyme polymerase in the RNA world may be circumvented: RNA could be copied by nonenzymatic chemistry alone until the advent of protein polymerases.

We suggest that the identification of a plausible source of chemical energy that could drive the re-activation of monomers and oligomers following hydrolytic loss of the leaving group is a key missing component in efforts to reconstitute nonenzymatic RNA replication. A chemical environment that could maintain a fully activated pool of substrates, and avoid accumulation of the inhibitory byproducts resulting from hydrolysis, might be sufficient to drive high yielding and complete copying of longer RNA templates in one pot, thus setting the stage for the emergence of Darwinian evolution.

## Materials and methods

### Chemicals

Guanosine 5′-monophosphate was purchased as the free acid from Santa Cruz Biotechnology (Dallas, TX). 2-thiouridine-5′-phosphoro-2methylimidazolide and 2′,3′-diacetyl-nucleosides were purchased from ChemGenes (Wilmington, MA). Phosphoramidite nucleotides and all other oligonucleotide synthesis reagents were purchased from either ChemGenes or Bioautomation (Plano, TX). Tris(hydroxymethyl)aminomethane (Tris)-HCl, ThermoScript reverse transcriptase as well as reaction buffer and NTPs were purchased from Thermo Fisher Scientific (Waltham, MA). DNA and RNA primers and were purchased from Integrated DNA Technologies (Coralville, IA). All oligonucleotide sequences are listed below. All other chemicals were purchased from Sigma Aldrich Corporation (St. Louis, MO).

### Gel electrophoresis

Gels were prepared using the SequaGel – UreaGel system from National Diagnostics (Atlanta, GA). Gels were prepared to 20% acrylamide and scanned using a Typhoon Scanner 9410 (GE Healthcare, Little Chalfont, Buckinghamshire, UK).

### Activated monomer and oligomer synthesis

RNA oligonucleotides were prepared by standard phosphoramidite oligonucleotide synthesis using a MerMade 6 DNA/RNA synthesizer (Bioautomation, Plano, TX) or starting with 2′,3′-diacetyl nucleosides using standard manual coupling procedures (*Beaucage and Caruthers, 1981*). 5′-phosphates were installed using bis-cyanoethyl-N,N-diisopropyl phosphoramidite from ChemGenes by standard phosphoramidite coupling chemistry.

Mononucleotide monophosphates and oligonucleotide monophosphates were activated using a modified published protocol (*Joyce et al., 1984*). As an example, 2-MeImpAGC (the 2-methylimidazolide of the trimer 5′-phosphoro-AGC) was synthesized by first dissolving 5 mg (5 μmole) of 5′-phosphoro-AGC in 1 mL of dimethyl sulfoxide (DMSO). To that mixture 160 mg (2 mmoles) of 2-methylimidazole, 56 mg triphenylphosphine (260 μmole), 64 mg 2,2′-dipyridyldisulfide (290 μmole)

and 40 µL triethylamine (550 µmole) were added. After stirring overnight at room temperature, the mixture was precipitated in 10 mL of a 400:250:30:1 mixture of acetone:diethylether:triethylamine: saturated solution of $NaClO_4$ in acetone. The precipitate was pelleted by centrifugation (3000 rpm, 5 min) and washed twice with a 1:1 mixture of acetone:diethylether and once with pure diethylether. After decanting the solvent, the pellet was dried under vacuum, resuspended in deionized water and purified by high-performance liquid chromatography (HPLC) over a $C_{18}$ column (Alltima $C_{18}$ 5 µm, Thermo Fisher, 250 × 10 mm) with 25 mM triethylammonium bicarbonate (TEAB) pH adjusted to 7.5 in 2% (v/v) acetonitrile for mobile phase A and acetonitrile for mobile phase B over a gradient beginning at 100% A and falling to 80% A over 20 min with a flow rate of 3 mL/minute. The fraction containing the product was verified by electrospray ionization mass spectrometry (ESI/MS) in negative mode (m/z = 996), frozen in liquid nitrogen and lyophilized overnight to yield a white powder. The powder was dissolved in water and the concentration of the activated trimer was determined using a NanoDrop 2000c spectrophotometer (Thermo Fisher Scientific, Waltham, MA) and calculated assuming $\varepsilon_{260}$ = 34,170 $M^{-1}$ $cm^{-1}$ [29].

Monomer stocks were adjusted to pH 7 before adding into the reaction mixture.

## Primer extension reactions

Nonenzymatic RNA primer extension reactions exploring the effect of downstream nucleotides and oligonucleotides on 2-MeImpG addition were conducted under the following conditions: 200 mM $MgCl_2$, 200 mM sodium N-cyclohexyl-2-aminoethanesulfonic acid (CHES) pH 9, 50 mM monomer, 1 mM trimer, 10 µM primer and 11 µM template. The primer sequence was the same for all reactions (6-carboxyfluorescein (FAM)-5′-GAC UGG-3′ for kinetics, cyanine 3 (Cy3)-5′-GCG UAG ACU GAC UGG-3′ for gels). The templates used in *Figure 1* had the following sequences:

1) 5′-AAC CCC CCA GUC -3′
2) 5′-G CUC CCA GUC -3′
3) 5′-AA CUC CCA GUC -3′

The templates in *Figure 1—figure supplement 2* had the following sequences:
For GGG:
5′-AA CCC CCA GUC -3′
For GCG:
5′-AA CGC CCA GUC -3′
For GAG:
5′-AA CUC CCA GUC -3′
For AGC:
5′-GCU GCU GCU GCU CCA GUC AGU CUA CGC-3′

The templates in *Figure 2* had the following sequences:
Part **a** lanes 1–4:
5′-G CUC CCA GUC -3′
Part **a** lane 5:
5′-GCC UCC CAG UC -3′
Part **b**:
5′-GCU **N** CCA GUC -3′

The templates in *Figure 3* had the same sequences as *Figure 2b*.

The template in *Figure 4* had the following sequence:
5′-CCC GCU ACC AGU C -3′

The template in *Figure 4—figure supplement 1* had the following sequence:
5′- GCU C GCU C GCU C GCU C CCAGUCAGUCUACGC -3′

The templates in *Figure 5* had the following sequences:
Part **a**:
5′-CCC GCU UCC AGU C -3′
Part **b**:
5′-AUC AAC CAG UC -3′

The template in *Figure 6* had the following sequence:
5′-biotin – AAC CCC GCG CCU CAU CAG CCA GUC -3′

The template in *Figure 6—figure supplement 1* had the following sequence:
5′ - GCG CCU CAU CAG CCA GUC -3′

Reaction progress was assessed by gel electrophoresis (20% acrylamide denaturing urea gel) after 10 min. Reaction rates were calculated by quantifying primer conversion to products using a Typhoon Scanner 9410 (GE Healthcare, Little Chalfont, Buckinghamshire, UK). Band intensities were quantified using ImageQuant TL software (GE Healthcare, Little Chalfont, Buckinghamshire, UK). The negative log of the fraction of unreacted primer was plotted against time, in hours. A linear regression was performed and the slope of the fit as plotted was reported as the pseudo-first order rate $k_{obs}$.

Primer extension reactions exploring the effect of different monomers and different oligomers on the rate of polymerization were conducted under the following conditions: 100 mM $MgCl_2$, 200 mM Tris-HCl pH 8, 25 mM monomer, 1 mM trimer, 10 µM primer and 11 µM template. The downstream helper oligomers were added in the following concentrations: 25 mM 2-MeImpA and 1 mM activated oligonuleotide (AG, AGC or AGGC). The reaction progress was assessed by gel electrophoresis (20% acrylamide denaturing urea gel) after 10 min. Reaction rates were calculated as above.

Primer extension reactions exploring the addition of multiple monomers and trimers in one reaction had the following conditions: 100 mM $MgCl_2$, 200 mM Tris-HCl pH 8, 25 mM monomers, 500 µM 2-methylimidazole-activated AGC, CGG and GGG, 100 µM activated GCG, 10 µM primer and 11 µM template. The reaction progress was assessed by gel electrophoresis (20% acrylamide denaturing urea gel) after 18 hr.

## Bead-immobilized reactions

30 µL of Dynabeads MyOne Streptavidin T1 (Invitrogen Dynal AS, Oslo, Norway) beads were prepared as specified by the manufacturer. Primer (Cy3 - 5′- GCG UAG ACU GAC UGG - 3′) and template (biotin - 5′- AAC CCC GCG CCU CAU CAG CCA GUC AGC UCU ACG C - 3′) were bound to the beads at 250 µM and 275 µM, respectively, in 2 M NaCl, 1 mM EDTA and 10 mM Tris-HCl pH 7. Monomer additions were conducted under the following conditions: 100 mM $MgCl_2$, 200 mM Tris-HCl pH 8, 25 mM monomer and 2 mM trimer. The beads were washed between steps in 2 M NaCl, 1 mM EDTA and 10 mM Tris-HCl pH 7. For steps 2, 5 and 7, 50 mM monomer and 4 mM trimer were added. Each step was monitored by gel electrophoresis until it reached >75% yield. Steps varied in length from 30 min to 22 hr; the total time required was 150 hr.

## Hammerhead ribozyme cleavage assay

10% of the product of the bead-immobilized reactions was washed twice in 100 µL water containing 0.01% (v/v) Tween 20. 174 µM DNA complement was added in 0.01% (v/v) Tween 20 and the beads were incubated at 95°C for 1 min. The solution was removed from the beads and 2 units of DNaseI along with DNaseI buffer (NEB) were added. Finally, the reaction was incubated at room temperature with 200 mM $MgCl_2$, 200 mM Tris-HCl pH 8 and 250 nM Cy5 labeled HH1 RNA (Cy5 - 5′ - CGC GCC GAA ACA CCG UGU CCC AGU C - 3′) for 20 hr. The products of both the bead immobilized reactions and the hammerhead reaction were analyzed by gel electrophoresis (20% acrylamide denaturing urea gel).

## Acknowledgements

We would like to thank Christian Hentrich, Aaron Engelhart, Ben Heuberger, Matthew Powner, Tony Jia, Alison Liou, Dipti Jasrasaria, Jeffrey Bessen, Alix Chan, Fred Rubino, Daniel Strassfeld and Holly Rees for helpful discussions and their contribution to this work. This work was supported in part by a grant (290363) from the Simons Foundation to JWS. NP was supported by a National Science Foundation Graduate Research Fellowship under Grant No. (DGE1144152). JWS is an Investigator of the Howard Hughes Medical Institute.

## Additional information

### Funding

| Funder | Grant reference number | Author |
| --- | --- | --- |
| Simons Foundation | 290363 | Noam Prywes<br>J Craig Blain<br>Francesca Del Frate<br>Jack W Szostak |
| National Science Foundation | DGE1144152 | Noam Prywes |

The funders had no role in study design, data collection and interpretation, or the decision to submit the work for publication.

### Author contributions

NP, Conception and design, Acquisition of data, Analysis and interpretation of data, Drafting or revising the article; JCB, Conception and design, Analysis and interpretation of data; FDF, Conception and design, Acquisition of data, Analysis and interpretation of data; JWS, Conception and design, Analysis and interpretation of data, Drafting or revising the article

### Author ORCIDs

Noam Prywes, http://orcid.org/0000-0003-2526-2392

J Craig Blain, http://orcid.org/0000-0001-5664-4065

Francesca Del Frate, http://orcid.org/0000-0003-2782-1290

Jack W Szostak, http://orcid.org/0000-0003-4131-1203

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
