## [Decision Letter]

Thank you for submitting your article "Catalysis of nonenzymatic template copying by activated oligonucleotides allows synthesis of an active RNA enzyme" for consideration by *eLife*. Your article has been reviewed by three peer reviewers, and the evaluation has been overseen by Timothy Nilsen, Reviewing Editor and John Kuriyan as the Senior Editor. The following individuals involved in review of your submission have agreed to reveal their identity: Gerald Joyce (Reviewer #2); Christopher Switzer (Reviewer #3).

The reviewers have discussed the reviews with one another and the Reviewing Editor has drafted this decision to help you prepare a revised submission.

Here Szostak and coworkers describe a major advance in the non-enzymatic template-directed polymerization of RNA, achieving dramatically faster reaction rates and much broader sequence generality compared to what has been achieved over the past 40 years of investigation. The authors uncover and exploit a heretofore unrecognized aspect of the templated polymerization of nucleoside 5'-phospho-2-methylimidazolides, involving catalysis by the nucleotide at the +1 position relative to the reacting nucleotide. Remarkably, the same 2-methylimidazole moiety that is the leaving group on the reacting nucleotide also has a catalytic effect at the +1 position. Furthermore, this catalytic effect can be provided by an adjacent oligonucleotide that has the same modification, thereby ensuring template occupancy of the adjacent catalyst. Employing a set of trinucleotide catalysts that cover successive reading frames, it now is possible to add sequentially all four nucleotides in good yield. Furthermore, if the trinucleotide catalysts are provided sequentially rather than as a mixture, even longer sequences can be synthesized, including 12 nucleotides that constitute a portion of the hammerhead ribozyme.

All of the referees felt that this work was an important contribution to nucleic acid chemistry and origins of life research.

Despite their enthusiasm all of the reviewers felt that the paper could be improved and each has spelled out a number of points that should be addressed via revision. In particular, all thought the statement in the manuscript that reacting nucleotides are the same for reactions 1 and 3 in Figure 1 needs to be addressed given that Figure 1 indicates the reacting nucleotides are different, G/G and G/A, respectively, and given that these experiments are together a key test of the "hierarchical assembly model." Please address this point as well as the other comments as thoroughly as possible.

Reviewer #1:

A long-standing question in origins research, and in the efforts to test/support the RNA World hypothesis specifically, is how RNA monomers (i.e., nucleotides) could have polymerized into polymeric RNA. This was the focus of the works of Orgel, Ferris, and several others and still remains unsatisfactorily solved. The current paper proposes a possible mechanism that partially addresses some of the chemical problems inherent to abiotic polymerization of nucleotides, namely that certain nucleotides (e.g., A and U) are recalcitrant towards non-enzymatic polymerization, and that in general such polymerization is slow, to the point of being ineffectual in the face of the back reaction (hydrolysis).

Here, the authors provide data that nucleotide activation can serve not only the canonical activation role that provides the free energy for polymerization, but also a catalytic role in that "downstream" activating groups can actually boost the rates up "upstream" ligation (polyermization) reactions. Specifically, the authors show that helper activated trinucleotide oligomers are particularly effective in enhancing rates of polymerization. In the extreme, they show that an RNA of sufficient length to be a catalyst itself, i.e., a ribozyme, can be made using this methodology, albeit with some help.

When I started reading the manuscript I immediately ran into a conceptual flaw that temporarily tempered my enthusiasm for the work. In the first paragraph of the Results, the authors use an (apparent) observation that, when activated with a specific "high-energy" leaving group (2-methylimidizole), the efficiency of mononucleotide polymerization is greater than that of oligonucleotide ligation. This would be somewhat unexpected because of several reasons including a perceived enhanced stability of the oligo on the template strand during the ligation reaction, compared to the stability of a single nucleotide on the template strand. But to show this, the authors compare the reactions #1 and #3 depicted in Figure 1. The accompanying text states that, "[…]the reacting nucleotides are the same in both cases: a G at the 3´ end of the primer and a G monomer or a G at the 5´ end of an oligonucleotide." But this is *not* the case: the oligomer in reaction #3 has an A on its 5´ end. This is particularly problematic because the authors make a big point about the lowered ability of A nucleotides to participate in ligation (or polymerization; the chemistry is the same) reactions compared to G. At first I thought it might be just a typo, but the same 5´-A appears in all of the subsequent helper trinucleotide oligomers. Unless I am reading this wrong, it appears as though the rationale for the rest of the project was thus based on a flawed premise.

Nevertheless, the balance of the manuscript did serve to convince me of the main result: that the activating group, and especially when placed on the 5´ ends of helper trinucleotides, can greatly facilitate monomer polymerization. This is indeed a significant result that should ultimately be meritorious of publication. I am always troubled by the artificiality of such imidazole-based activating groups, and thus I would ask the authors to comment on this more explicitly, and perhaps try other such activating groups. Would this work with a triphosphate or even with a 2´-3´-cyclic phosphate, the later has a clear prebiotic route to its synthesis as shown by Sutherland?

Obviously the flaw mentioned above about the G vs. A should also be addressed; this could be done for example by reformulating the rationale for the study (or showing me how I misunderstand the Figure/text correspondence).

On another point, the mechanism suggested by these data is reminiscent of "substrate assisted catalysis", which at one point was posited to be involved in the mechanism of RNA-directed peptide-bond formation in the PTC of the ribosome. This could be discussed.

The fact that a trinucleotide emerges as the best catalyst (and in fact I disagree with the characterization in the third paragraph of Results that the reaction rate stays "constant" with the tetranucleotide; from Figure 2 there appears to be a significant drop) is intriguing. Could this be related to the thermodynamic "sweet spot" for iterative information transfer, as seen in the 3-bp codon-anticodon interaction?

I wonder what the effect of hot/cold or wet/dry cycles would have on polymerization as assisted in the proposed scheme.

My last major point is that the authors, in showing these data and proposing this chemistry, miss an opportunity to note that perhaps the elusive RNA-dependent RNA polymerase ribozyme (RNA replicase ribozyme) never existed. With substrate-assisted non-enzymatic template-directed RNA replication, one could envisage moving from a abiotic RNA monomer world to a protein world in a manner that obviates the need for an RNA replicase ribozyme.

Reviewer #2:

Prywes et al. describe a major advance in the non-enzymatic template-directed polymerization of RNA, achieving dramatically faster reaction rates and much broader sequence generality compared to what has been achieved over the past 40 years of investigation. The authors uncover and exploit a heretofore unrecognized aspect of the templated polymerization of nucleoside 5'-phospho-2-methylimidazolides, involving catalysis by the nucleotide at the +1 position relative to the reacting nucleotide. Remarkably, the same 2-methylimidazole moiety that is the leaving group on the reacting nucleotide also has a catalytic effect at the +1 position. Furthermore, this catalytic effect can be provided by an adjacent oligonucleotide that has the same modification, thereby ensuring template occupancy of the adjacent catalyst. Employing a set of trinucleotide catalysts that cover successive reading frames, it now is possible to add sequentially all four nucleotides in good yield. Furthermore, if the trinucleotide catalysts are provided sequentially rather than as a mixture, even longer sequences can be synthesized, including 12 nucleotides that constitute a portion of the hammerhead ribozyme.

This manuscript is an important contribution to nucleic acid chemistry and origins-of-life research. It will have appeal to a broad audience of chemists and biologist who are interested in the origin of genetic replication systems. The manuscript is highly suitable for publication in *eLife*, subject to the following modifications.

1) In the first paragraph of the Results. A reference should be added to the 1999 paper by James and Ellington (Origins Life Evol. Biosph. 29:375-390), which first described a prebiotic scenario for template copying based on ligation of short oligonucleotides.

2) In the first paragraph of the Results. The description of the comparative addition of either a mono- or oligonucleotide to a template-bound primer is confusing because it states that the reacting nucleotides are the same, but the referenced reactions in Figure 1 involve either addition of G (reaction 1) or ligation of AGC (reaction 3). Presumably the ligation reaction involved addition of GCG, but was omitted from the figure.

3) At the end of the Results. It is an exaggeration to say that "an active ribozyme was synthesized nonenzymatically". Of the 37 nucleotides that constitute the catalytic complex, only 12 were synthesized non-enzymatically, with 6 provided by the primer and 19 contained within the substrate portion of the complex. Although Wochner et al. (Science 332:209-212, 2011) used similar artifice to "synthesize" a hammerhead ribozyme using a polymerase ribozyme, the present manuscript should strive for a higher standard. The title of the manuscript is similarly misleading and should be changed from "[…]Allows Synthesis of an Active RNA Enzyme" to something like "[…]Allows Synthesis of a broad range of RNA Sequences".

4) In the second paragraph of the Discussion. Further explanation is needed regarding what is meant by "the lack of a leaving group one base downstream of the site of primer extension". The meaning is clear to a specialist, but a non-specialist needs to be reminded that the catalytic 2-methylimidazole is not present at the second nucleotide position of the helper trinucleotide (and in any case would not be a "leaving group") because this is an unmodified phosphodiester within the 5´-activated trinucleotide.

5) In the second paragraph of the Discussion. The authors are being too coy when referring to the possibility of an imidazolium intermediate that is the basis for the catalytic effect. If they have evidence for such an intermediate in another study, then it should be referred to as "unpublished data" or "manuscript in press", as appropriate.

6) In the third paragraph of the Discussion. It is an exaggeration to say that the non-enzymatic polymerization reaction reported here compares favorably in terms of efficiency with the tC19Z polymerase ribozyme reported by Wochner et al. (Science 332:209-212, 2011). While the rates are comparable, the efficiencies are not even close, especially in batch reactions where Prywes et al. generate no more than 7mers and Wochner et al. generate products (albeit of an especially favorable sequence) of more than 90 nucleotides.

7) Figure 3. The heat map depiction of fidelities does not do justice to the data, which would be better represented as actual numbers.

8) Figure 4. The lower right panel of the figure is confusing. Presumably it corresponds to the data shown in Figure 4, although this is not clear from the figure legend. Adding to the confusion are the lack of labels along the vertical axis of the gel, the triangle presumably representing increasing time, and the non-standard labeling convention. The figure should be redone more thoughtfully.

Reviewer #3:

The manuscript reports nonenzymatic template directed copying reactions of activated monomers promoted by activated oligonucleotides. The system described enabled the template directed synthesis of an active hammerhead ribozyme. This work is the first to report the nonenzymatic synthesis of a catalytic RNA using activated monomers. Prebiotic chemists have sought the means to simply and robustly copy all four RNA monomers for decades, with mostly limited success, for A and U in particular. Moreover, it has become increasingly apparent that the limitations imposed by A and U are nuanced, extending beyond simply diminished binding affinities. In keeping with the subtle nature of these systems, the authors have taken a known, but poorly understood, feature of 2-Me-imidazolide activated nucleoside phosphate monomers noted by Orgel, and applied it in a heretofore unanticipated manner. The authors surprising initially discovered that the rate of 2-MeImpN oligomerization exceeded the rate of activated trinucleotide ligation by nearly 100-fold, despite the greater template binding affinity of the trimer. On further investigation it became evident that consecutive 2-MeImp groups in subsequent monomers led to the enhanced rate, and, ultimately, the unusual discovery that an 2-MeImp activated trinucleotide enhanced the rate of monomer addition due to the neighboring 2-MeImp group. Thus, while Orgel had earlier shown that at least two consecutive 2-MeImpN monomers were required for efficient template directed synthesis, it was not possible to anticipate the effectiveness of the presently reported system for a variety of reasons, not least because it involves an activated trinucleotide rather than activated monomers, and historically there appeared little hope for efficient synthesis with A and U, consecutive 2-MeImp activating groups notwithstanding. The prospects of simplifying the reported system even further in future work are very exciting for the field.

Introduction, first and second paragraphs. It is awkward to offer template immobilization as a shortcoming, but then promote its use shortly afterwards, even though template immobilization in the first instance is one of several shortcomings.

Results, first paragraph. This sentence states the reacting nucleotides are the same in both reactions 1 and 3 in Figure 1 by including a G monomer or a G at the 5'-end of an oligonucleotide. However, the oligonucleotide shown for reaction 3 has an A at the 5'-end.

Results, fifth paragraph. It would be extremely interesting to also know what effect the 2-thio U or rT has on the rate of the G:(s)U/rT mismatch reaction.

Results, last paragraph. This should reference Figure 6.

Discussion, second paragraph. In Wu and Orgel, 1992, Orgel notes in the Discussion that simple steric effects are unlikely, suggesting instead that the neighboring 2-MeIm might act as an acid/base catalyst. I believe this section should be revised to reflect this fact. Nevertheless, the suggestion by the authors of covalent bond formation is new and interesting.

---

## [Author Response]

Despite their enthusiasm all of the reviewers felt that the paper could be improved and each has spelled out a number of points that should be addressed via revision. In particular, all thought the statement in the manuscript that reacting nucleotides are the same for reactions 1 and 3 in Figure 1 needs to be addressed given that Figure 1 indicates the reacting nucleotides are different, G/G and G/A, respectively, and given that these experiments are together a key test of the "hierarchical assembly model." Please address this point as well as the other comments as thoroughly as possible.

We would like to thank the Editor and all of the reviewers for alerting us to this critical oversight, which resulted from replacing an earlier experiment with more recent data. We have now included a set of experiments that provide a direct comparison between the addition of multiple G monomers and an oligonucleotide that begins with G. Figure 1 has been changed to incorporate this new data, and a new figure supplement, Figure 1—figure supplement 2, has been added that includes additional oligonucleotide ligation reactions. We believe that the newly added data addresses this concern fully.

*Reviewer #1:*

When I started reading the manuscript I immediately ran into a conceptual flaw that temporarily tempered my enthusiasm for the work. In the first paragraph of the Results, the authors use an (apparent) observation that, when activated with a specific "high-energy" leaving group (2-methylimidizole), the efficiency of mononucleotide polymerization is greater than that of oligonucleotide ligation. This would be somewhat unexpected because of several reasons including a perceived enhanced stability of the oligo on the template strand during the ligation reaction, compared to the stability of a single nucleotide on the template strand. But to show this, the authors compare the reactions #1 and #3 depicted in Figure 1. The accompanying text states that, "[…]the reacting nucleotides are the same in both cases: a G at the 3´ end of the primer and a G monomer or a G at the 5´ end of an oligonucleotide." But this is not the case: the oligomer in reaction #3 has an A on its 5´ end. This is particularly problematic because the authors make a big point about the lowered ability of A nucleotides to participate in ligation (or polymerization; the chemistry is the same) reactions compared to G. At first I thought it might be just a typo, but the same 5´-A appears in all of the subsequent helper trinucleotide oligomers. Unless I am reading this wrong, it appears as though the rationale for the rest of the project was thus based on a flawed premise.

Nevertheless, the balance of the manuscript did serve to convince me of the main result: that the activating group, and especially when placed on the 5´ ends of helper trinucleotides, can greatly facilitate monomer polymerization. This is indeed a significant result that should ultimately be meritorious of publication. I am always troubled by the artificiality of such imidazole-based activating groups, and thus I would ask the authors to comment on this more explicitly, and perhaps try other such activating groups. Would this work with a triphosphate or even with a 2´-3´-cyclic phosphate, the later has a clear prebiotic route to its synthesis as shown by Sutherland?

We use the 2-methylimidazole activating group because it is the best model system available for the study of nonenzymatic template copying, irrespective of its its prebiotic relevance. We have added a sentence to this effect to the manuscript: “Nucleotides activated with 2-methylimidzole on the 5′-phosphate have been used extensively to model nonenzymatic template copying (Weimann et al., 1968; Wu and Orgel, 1992; Deck, Jauke and Richert, 2011), irrespective of their prebiotic plausibility, and were used in all experiments reported here.”

We and others do think about possible prebiotic routes to this and other activating groups, and this is the subject of ongoing research in several labs. As far as we are aware, no nonenzymatic polymerization of nucleoside triphosphates or 2´-3´-cyclic phosphates has ever been reported, although the corresponding oligonucleotide ligation reactions have been studied.

Obviously the flaw mentioned above about the G vs. A should also be addressed; this could be done for example by reformulating the rationale for the study (or showing me how I misunderstand the Figure/text correspondence).

See response above to Editor/Reviewer #1.

On another point, the mechanism suggested by these data is reminiscent of "substrate assisted catalysis", which at one point was posited to be involved in the mechanism of RNA-directed peptide-bond formation in the PTC of the ribosome. This could be discussed.

While this is an interesting suggestion, we think that our findings are not really an example of substrate assisted catalysis, because the substrate (the activated monomer) is distinct from the catalyst (the activated downstream nucleotide or oligonucleotide).

The fact that a trinucleotide emerges as the best catalyst (and in fact I disagree with the characterization in the third paragraph of Results that the reaction rate stays "constant" with the tetranucleotide; from Figure 2 there appears to be a significant drop) is intriguing. Could this be related to the thermodynamic "sweet spot" for iterative information transfer, as seen in the 3-bp codon-anticodon interaction?

We thank the reviewer for pointing out an intriguing possibility and a possibly important parallel with the codon-anticodon interaction. There is indeed a drop from trinucleotide to tetranucleotide catalysis, although we suspect that this drop is within the variance due to different sequence contexts and would appear insignificant if additional trimer and tetramer sequences were tested as helpers. It is certainly possible that trimers might be optimal for continued primer extension, which requires that the helper oligo bind, catalyze, then dissociate. However, we have not yet carried out the experiments needed to test this hypothesis, and in the absence of evidence we would prefer not to speculate on this issue.

We have edited the manuscript to read (Results, third paragraph): “approximately constant”

I wonder what the effect of hot/cold or wet/dry cycles would have on polymerization as assisted in the proposed scheme.

We have attempted to optimize the temperature of the reaction and have found that within a window of approximately 10-40 degrees Celsius there is no noticeable effect on the rate while outside of that window the reaction slows down precipitously. Thermocycling was similarly ineffective in improving the reaction product distribution. We did not test the effect of drying and rewetting our samples. Such experiments may be fruitful but are unfortunately outside of the scope of this study.

My last major point is that the authors, in showing these data and proposing this chemistry, miss an opportunity to note that perhaps the elusive RNA-dependent RNA polymerase ribozyme (RNA replicase ribozyme) never existed. With substrate-assisted non-enzymatic template-directed RNA replication, one could envisage moving from a abiotic RNA monomer world to a protein world in a manner that obviates the need for an RNA replicase ribozyme.

We have edited the text to reflect this possibility by adding the following statement: “In fact, if the rates and fidelities of nonenzymatic RNA polymerization were sufficiently high, the need for a ribozyme polymerase in the RNA world might be circumvented: RNA could be copied by nonenzymatic chemistry alone until the advent of protein polymerases.”

*Reviewer #2:*

*This manuscript is an important contribution to nucleic acid chemistry and origins-of-life research. It will have appeal to a broad audience of chemists and biologist who are interested in the origin of genetic replication systems. The manuscript is highly suitable for publication in eLife, subject to the following modifications.*

1) In the first paragraph of the Results. A reference should be added to the 1999 paper by James and Ellington (Origins Life Evol. Biosph. 29:375-390), which first described a prebiotic scenario for template copying based on ligation of short oligonucleotides.

We have added this reference.

2) In the first paragraph of the Results. The description of the comparative addition of either a mono- or oligonucleotide to a template-bound primer is confusing because it states that the reacting nucleotides are the same, but the referenced reactions in Figure 1 involve either addition of G (reaction 1) or ligation of AGC (reaction 3). Presumably the ligation reaction involved addition of GCG, but was omitted from the figure.

See response above to Editor/Reviewer #1.

3) At the end of the Results. It is an exaggeration to say that "an active ribozyme was synthesized nonenzymatically". Of the 37 nucleotides that constitute the catalytic complex, only 12 were synthesized non-enzymatically, with 6 provided by the primer and 19 contained within the substrate portion of the complex. Although Wochner et al. (Science 332:209-212, 2011) used similar artifice to "synthesize" a hammerhead ribozyme using a polymerase ribozyme, the present manuscript should strive for a higher standard. The title of the manuscript is similarly misleading and should be changed from "[…]Allows Synthesis of an Active RNA Enzyme" to something like "[…]Allows Synthesis of a broad range of RNA Sequences".

We thank the reviewer for holding us to a higher standard and have edited the text to reflect this as follows: “[…]thus demonstrating that *a significant portion of* an active ribozyme was synthesized nonenzymatically.”

We have changed the title to: Nonenzymatic copying of RNA templates containing all four letters is catalyzed by activated oligonucleotides

4) In the second paragraph of the Discussion. Further explanation is needed regarding what is meant by "the lack of a leaving group one base downstream of the site of primer extension". The meaning is clear to a specialist, but a non-specialist needs to be reminded that the catalytic 2-methylimidazole is not present at the second nucleotide position of the helper trinucleotide (and in any case would not be a "leaving group") because this is an unmodified phosphodiester within the 5´-activated trinucleotide.

We have edited the text to improve clarity by adding the following: “[…]in this case the downstream position corresponds to the second monomer in the oligonucleotide which necessarily has no leaving group as it is linked to the upstream monomer by a phosphodiester linkage.”

5) In the second paragraph of the Discussion. The authors are being too coy when referring to the possibility of an imidazolium intermediate that is the basis for the catalytic effect. If they have evidence for such an intermediate in another study, then it should be referred to as "unpublished data" or "manuscript in press", as appropriate.

We have added (Discussion, second paragraph) a reference to recent work describing such a covalent adduct, and a reference in the text to our ongoing work, “T. Walton and J.W. Szostak, unpublished data.”

6) In the third paragraph of the Discussion. It is an exaggeration to say that the non-enzymatic polymerization reaction reported here compares favorably in terms of efficiency with the tC19Z polymerase ribozyme reported by Wochner et al. (Science 332:209-212, 2011). While the rates are comparable, the efficiencies are not even close, especially in batch reactions where Prywes et al. generate no more than 7mers and Wochner et al. generate products (albeit of an especially favorable sequence) of more than 90 nucleotides.

We have edited the text (Discussion, third paragraph) to emphasize that our rates and efficiencies are close to those obtained by Wochner et al. on arbitrary RNA templates, as opposed to the specific sequence on which the Wochner polymerase was trained.

7) Figure 3. The heat map depiction of fidelities does not do justice to the data, which would be better represented as actual numbers.

We have added the actual numbers into Figure 3—figure supplement 1. We believe that the heat map provides a helpful visual summary of the data that will be more useful to the general reader, and therefore we have retained this figure in the main paper.

8) Figure 4. The lower right panel of the figure is confusing. Presumably it corresponds to the data shown in Figure 4, although this is not clear from the figure legend. Adding to the confusion are the lack of labels along the vertical axis of the gel, the triangle presumably representing increasing time, and the non-standard labeling convention. The figure should be redone more thoughtfully.

We thank the reviewer for the helpful comment and have edited Figure 4 for clarity. We have added labels to the vertical axis of the gels; we have removed the triangle representing time and added numbers for reaction times.

*Reviewer #3:*

Introduction, first and second paragraphs. It is awkward to offer template immobilization as a shortcoming, but then promote its use shortly afterwards, even though template immobilization in the first instance is one of several shortcomings.

We have deleted the sentence on template immobilization.

Results, first paragraph. This sentence states the reacting nucleotides are the same in both reactions 1 and 3 in Figure 1 by including a G monomer or a G at the 5'-end of an oligonucleotide. However, the oligonucleotide shown for reaction 3 has an A at the 5'-end.

See response above to Editor/Reviewer #1.

Results, fifth paragraph. It would be extremely interesting to also know what effect the 2-thio U or rT has on the rate of the G:(s)U/rT mismatch reaction.

We are currently pursuing additional experiments related to the fidelity of nonenzymatic primer extension, using both MS and deep sequencing methods to analyze reactions in which the different monomers can compete with each other for binding to the template. These studies, which will be published separately, will address the interesting question of how the 2-thio pyrimidines will affect fidelity.

Results, last paragraph. This should reference Figure 6.

We have edited the text to correct this mistake.

*Discussion, second paragraph. In Wu and Orgel, 1992, Orgel notes in the Discussion that simple steric effects are unlikely, suggesting instead that the neighboring 2-MeIm might act as an acid/base catalyst. I believe this section should be revised to reflect this fact. Nevertheless, the suggestion by the authors of covalent bond formation is new and interesting.*

We have edited the text in accordance with this suggestion as follows: “[…]though the authors ultimately concluded that such subtle effects are not likely to adequately explain the strength of the interaction. Wu et al. also suggested that the catalytic effect could be due to acid-base catalysis.”